

# Exploring mechanisms that affect coral cooperation: symbiont transmission mode, cell density and community composition

Carly D. Kenkel[1] and Line K. Bay[2]

[1] Department of Biological Sciences, University of Southern California, Los Angeles, CA, USA
[2] Australian Institute of Marine Science, Townsville, QLD, Australia

Corresponding author
Carly D. Kenkel, ckenkel@usc.edu

## ABSTRACT

The coral symbiosis is the linchpin of the reef ecosystem, yet the mechanisms that promote and maintain cooperation between hosts and symbionts have not been fully resolved. We used a phylogenetically controlled design to investigate the role of vertical symbiont transmission, an evolutionary mechanism in which symbionts are inherited directly from parents, predicted to enhance cooperation and holobiont fitness. Six species of coral, three vertical transmitters and their closest horizontally transmitting relatives, which exhibit environmental acquisition of symbionts, were fragmented and subjected to a 2-week thermal stress experiment. Symbiont cell density, photosynthetic function and translocation of photosynthetically fixed carbon between symbionts and hosts were quantified to assess changes in physiological performance and cooperation. All species exhibited similar decreases in symbiont cell density and net photosynthesis in response to elevated temperature, consistent with the onset of bleaching. Yet baseline cooperation, or translocation of photosynthate, in ambient conditions and the reduction in cooperation in response to elevated temperature differed among species. Although *Porites lobata* and *Galaxea acrhelia* did exhibit the highest levels of baseline cooperation, we did not observe universally higher levels of cooperation in vertically transmitting species. *Post hoc* sequencing of the *Symbiodinium* ITS-2 locus was used to investigate the potential role of differences in symbiont community composition. Interestingly, reductions in cooperation at the onset of bleaching tended to be associated with increased symbiont community diversity among coral species. The theoretical benefits of evolving vertical transmission are based on the underlying assumption that the host-symbiont relationship becomes genetically uniform, thereby reducing competition among symbionts. Taken together, our results suggest that it may not be vertical transmission *per se* that influences host-symbiont cooperation, but genetic uniformity of the symbiont community, although additional work is needed to test this hypothesis.

# INTRODUCTION

Cooperation between species has played a fundamental role in the evolution and diversification of life (*Friesen & Jones, 2013*; *Kiers & West, 2015*; *Maynard Smith &*

*Szathmary, 1995*; *Moran, 2006*). In the case of reef-building corals, the intracellular symbiosis between dinoflagellates in the genus *Symbiodinium* and a calcifying Cnidarian host forms the basis of one of the most biodiverse and productive ecosystems on the planet (*Hatcher, 1988*; *Knowlton et al., 2010*). The process of host calcification, which builds the three dimensional structure of the reef, is largely powered by symbiont primary productivity (*Roth, 2014*). However, climate change and other anthropogenic processes threaten reefs because of the sensitivity of the coral-dinoflagellate symbiosis to environmental stress (*Hoegh-Guldberg et al., 2007*; *Hughes et al., 2003*), indicating that host-symbiont cooperation is not stable over ecological timescales.

Recent work has suggested that a transition to parasitism may precipitate the breakdown of the host-symbiont relationship (*Baker et al., 2018*), but this is not a unique feature of the coral symbiosis. Across taxa, mutualisms are better defined as a spectrum that ranges from negative parasitic interactions to mutually beneficial symbioses, both within the context of a focal inter-species interaction and when comparing relationships across taxa (*Doebeli & Knowlton, 1998*; *Lesser, Stat & Gates, 2013*; *Nowak, Bonhoeffer & May, 1994*; *Sachs, Essenberg & Turcotte, 2011*). Less well-resolved, particularly for the coral-*Symbiodinium* symbiosis, are the mechanisms that promote and maintain positive interactions between partners (*Lesser, Stat & Gates, 2013*; *Sachs & Wilcox, 2006*).

One major factor predicted to influence levels of cooperation is the mode of symbiont transmission (*Anderson & May, 1982*; *Ebert & Bull, 2003*). In corals as in other symbioses, two transmission modes predominate: symbionts can be acquired horizontally from the local environment, usually during a defined larval stage, or vertically from parents, typically through the maternal germ line (reviewed in *Bright & Bulgheresi (2010)*). Virulence theory predicts that horizontal transmission allows symbionts to adopt selfish strategies, potentially harmful to the host (*Bull, 1994*). A transition from horizontal to vertical transmission is predicted to align the reproductive interests of partners (via partner-fidelity feedback *sensu Sachs et al., 2004*) and optimize resource sharing to maximize holobiont (the combination of host and symbiont) fitness (*Ebert, 2013*; *Frank, 1994*; *Herre et al., 1999*).

Experimental manipulations of transmission mode in other systems have provided empirical support for a reduction in pathogen virulence under enforced vertical transmission scenarios (*Bull, Molineux & Rice, 1991*; *Dusi et al., 2015*; *Sachs & Wilcox, 2006*; *Stewart, Logsdon & Kelley, 2005*). Bacteriophages forced into vertical transmission evolved lower virulence and lost the capacity to transmit horizontally (*Bull, Molineux & Rice, 1991*). Similarly, *Symbiodinium microadriaticum* under an experimentally enforced horizontal transmission regime proliferated faster within their *Cassiopea* jellyfish hosts while reducing host reproduction and growth (*Sachs & Wilcox, 2006*). However, studies attempting to quantify the evolutionary consequences of natural shifts in transmission mode remain rare, with Herre's demonstration of a negative relationship between vertical transmission and virulence in nematodes that parasitize fig wasps a notable exception (*Herre, 1993*).

Reef-building corals are a potential system in which to study naturally occurring transitions in transmission mode in a mutualistic symbiosis. Corals (Cnidaria: Anthozoa: Scleractinia) are colonial animals that harbour intracellular populations of dinoflagellate algae in the genus *Symbiodinium*. This symbiosis is considered obligate as the breakdown of the relationship between host animals and their intracellular *Symbiodinium* algae, commonly known as coral bleaching, has major fitness consequences for both partners, and can be lethal (reviewed in *Brown (1997)*). This inter-species partnership is ancient (evolved ~250 Mya (*Stanley & Swart, 1995*), prolific (600+ coral species worldwide (*Daly et al., 2007*), and constitutes the foundation of one of the most bio-diverse ecosystems on the planet. The majority of coral species (~85%) acquire their *Symbiodinium* horizontally from the local environment in each generation (*Harrison & Wallace, 1990*). However, vertical transmission has independently evolved at least four times, such that both transmission strategies can be exhibited by different coral species within the same genus (*Baird, Guest & Willis, 2009b*; *Hartmann et al., 2017*).

We compared physiological components of cooperation and fitness proxies between horizontal and vertical transmitters (VTs) in a phylogenetically controlled design using three pairs of related coral species exhibiting different strategies: (1) *Galaxea acrhelia* (VT) and *Galaxea astreata* (horizontal transmitter, HT); (2) *Porites lobata* (VT) *and Goniopora columna* (HT); (3) *Montipora aequituberculata* (VT) and *Acropora millepora* (HT). Species comparisons were drawn from the same or sister genera and replicate comparisons from more distantly related clades (*Hartmann et al., 2017*). Species also represented a diversity of reproductive modes (e.g. broadcast spawner, brooder), sexual systems (e.g. hermaphroditic, gonochoric), and morphologies (e.g. massive, branching), and host different subclades of *Symbiodinium* (*Tonk et al., 2013*), such that mode of symbiont transmission was the only consistent difference between pairs (*Baird, Guest & Willis, 2009b*; *Franklin et al., 2012*; *Kerr, Baird & Hughes, 2011*). We quantified changes in symbiont cell density, photosynthetic function and translocation of photosynthetically fixed carbon between symbionts and hosts. We defined cooperation as the proportion of photosynthetically fixed carbon translocated to the host, while the degree of symbiont parasitism was calculated as the difference in the proportion of photosynthetically fixed carbon translocated to hosts between control and heat-treated samples at the end of the experiment, *sensu Baker et al. (2018)*. Bleaching, or the reduction in symbiont density in response to sustained thermal stress was used as a proxy for holobiont fitness. While we observed differences in host-symbiont cooperation, both at a baseline level and during the onset of bleaching, vertically transmitting species did not exhibit universally elevated levels of cooperation. Additional *post hoc* analysis of *Symbiodinium* ITS-2 diversity among coral species was therefore used to investigate whether symbiont community composition could better explain physiological trait patterns. Symbiont community composition did not explain a significant portion of the variation in physiological components of cooperation or fitness proxies; however, diversity tended to be associated with the degree of symbiont parasitism at the onset of bleaching, suggesting that reduced genetic diversity of symbionts, rather than vertical transmission *per se*, may influence host-symbiont cooperation.

## MATERIALS AND METHODS

### Coral collection and acclimation

Fragments from 60 unique coral colonies, ~20 cm in diameter, were collected from reefs on the Central GBR from the 8–22 April 2015 under the Great Barrier Reef Marine Park Authority permits G12/35236.1 and G14/37318.1, prioritizing collection of focal species pairs by transmission mode from the same reef environment, including location and depth. A total of 10 corals of each species, *Galaxea acrhelia* and *Galaxea astreata* were collected from Davies Reef (18°49.816′, 147°37.888′, 11 April 2015), 10 corals each of *A. millepora* and *M. aequituberculata* were collected from Pelorus Island (18°33.358′, 146°30.276′, 18 April 2015) and 10 corals each of *Goniopora columna* and *P. lobata* were collected from Pandora Reef (18°48.778′, 146°25.593′, 21 April 2015) from depths of <10 m across all reefs, and from the same depth within each site. Corals were transported to the Australian Institute of Marine Science Sea Simulator facility and placed in shaded outdoor holding tanks with 0.2 μM filtered flow-through seawater (FSW, 27 °C, 150 μmol quanta m$^2$/s). Each individual coral colony was further cut into six replicates using a diamond blade band saw and these fragments were mounted on aragonite plugs using either super-glue or marine epoxy. On 1 May 2015, fragments were moved into indoor experimental aquaria, which consisted of 12 50-L treatment tanks fitted with 3.5 W Turbelle nanostream 6,015 pumps (Tunze, Penzberg, Germany) with filtered flow-through seawater (FSW, ~25 l/h) at 27 °C with Hydra 52 lights (Aqua Illumination, C2 Development, Inc., Ames, IA, USA) on a 12:12 light/dark cycle mimicking natural irradiance patterns, peaking at 130–160 μmol quanta m$^2$/s at 'midday', which actually occurred at 10:30 GMT +10. Tanks, coral racks and plugs were cleaned daily. Beginning on 7 May, at 20 mins before dark (16:30 GMT +10) every day, corals were fed *Artemia* naupli at a density of 1–1.5 naupli/5 ml and rotifers at a density of 1–3/ml.

### Experimental conditions and physiological trait measurements

Beginning on 28 May 2015, effective quantum yield (EQY) of *Symbiodinium* photosystem II was measured daily for all experimental fragments using a pulse amplitude modulated fluorometer (diving-PAM, Walz) fitted with a plastic fibre optic cable (Fig. S1). Measurements were made using factory settings with a measuring intensity of 12 and a gain of 5 and taken at peak light intensity (between 09:30 and 11:30 GMT +10). EQY values were used to guide the timing of the final sampling time point (Fig. S2), where a decline reflects an impact on the photosynthetic condition of the *Symbiodinium* (*Ralph, Larkum & Kuhl, 2005*) because we aimed to target the onset of the coral bleaching response rather than the end-point.

On 31 May 2015, temperatures in the heat treatment tanks were increased at a rate of 1 °C per day until temperatures reached 31 °C (day 4, Fig. S3). Sample time points occurred on day 2 (29 °C, 1 June), day 4 (31 °C, 3 June) and day 17 (31 °C, 16 June, Fig. S3). On each sampling day, one replicate fragment of each colony ($n = 10$) and species ($n = 6$) from each temperature treatment ($n = 2$; $n = 120$ total per sampling day) were used to measure net photosynthesis following the two-point method originally described and

validated for *A. millepora* in *Strahl et al. (2015)*. It is important to note that this method was not validated for additional species or under the experimental conditions used in the present study, nor were species specific photosynthesis-irradiance curves quantified to determine an appropriate saturating irradiance. Briefly, corals were incubated in enclosed 600 ml acrylic chambers at their respective treatment temperatures and light levels for 1.5 h. Chambers were placed onto custom built tables with rotating magnets, which served to power stir bars within each chamber to facilitate water mixing. For each run, four chambers without corals were used as blanks to account for potential changes in oxygen content due to the metabolic activity of other microorganisms in the seawater. For net photosynthesis measures, the $O_2$ concentration of the seawater in each chamber was measured at the end of the run using a hand-held dissolved oxygen meter (HQ30d, equipped with LDO101 IntelliCAL oxygen probe; Hach, Loveland, CO, USA). Values from blank chambers were subtracted from measures made in coral chambers and the subsequent rate of net photosynthesis was related to coral surface area, calculated in $\mu g\ O_2/cm^2/min$.

To measure the fraction of autotrophically derived carbon translocated to host animals, five colony fragments of each species from each treatment ($n = 30$ control, $n = 30$ heat) were placed into 18-L of FSW with a 5-W aquarium pump for circulation and $^{14}C$-bicarbonate (specific activity: 56 mCi/mmol) was added to a final concentration of 0.28 $\mu Ci/ml$. Corals were incubated for 5 h in experimental tanks which experienced the normal experimental irradiance profile from 09:00 to 14:00 GMT +10, rinsed with flow-through FSW for 1 h to remove remaining unfixed $^{14}C$ then snap frozen in liquid nitrogen.

Tissue was removed from snap frozen coral skeletons using an air gun and homogenized for 60 s using a Pro250 homogenizer (Perth Scientific Equipment, Perth, WA, Australia). A 300 $\mu$l aliquot of the tissue homogenate was fixed with 5% formalin in FSW and used to quantify *Symbiodinium* cell density. The average cell number was obtained from four replicate haemocytometer counts of a 1 $mm^3$ area and cell density was related to host protein content (as assessed below) and expressed as cells/mg host protein. Although surface area is commonly used as the normalization factor for *Symbiodinium* cell density, it has been recognized that areal abundance does not account for differences in host biomass (reviewed in *Cunning & Baker (2014)*). We found that normalization to soluble host protein more accurately reflected this difference in biomass, given that tissue thickness is significantly greater in *Goniopora columna* and not adequately accounted for by skeletal surface area normalization alone (Fig. S4). We determined the degree of bleaching, or the reduction in symbiont density in response to sustained thermal stress, by calculating the difference in symbiont cell densities between heat-treated and paired control samples following the full 17 days of experimental treatment and used this value as a proxy of holobiont fitness.

An additional 1 ml aliquot of holobiont homogenate was frozen at $-20\ ^\circ C$. The remaining homogenate was centrifuged for 2 min at 3,500 rcf to separate host and symbiont fractions and two ml of the host tissue slurry was frozen at $-20\ ^\circ C$. Total soluble protein was quantified for host tissue samples in duplicate using a colorimetric assay (Bio-Rad Protein Assay Kit II; Bio-Rad Laboratories, Inc., Hercules, California, USA)
following the manufacturer's instructions. Coral skeletons were rinsed with 5% bleach then dried at room temperature. Skeletal surface area was quantified using the single wax dipping method (*Veal et al., 2010*) and skeletal volume (used to standardize respirometry chamber volumes) was determined by calculating water displacement in a graduated cylinder.

Sample radioactivity was determined using a liquid scintillation counter (Tri-Carb 2810TR v2.12; Perkin Elmer, Waltham, MA, USA). Triplicate host and duplicate holobiont 300 µl tissue homogenate aliquots were mixed with 3.5 ml Ultima Gold XR liquid scintillation cocktail (Perkin Elmer, Waltham, MA, USA). Samples were temperature and light adapted for 1 h and then counted for 1.5 min using the default parameters. Counts per minute were converted to disintegrations per minute (DPM) using a standard curve derived from a $^{14}$C Ultima Gold Quench Standards Assay (Perkin Elmer, Waltham, MA, USA). Technical replicates were averaged. Host DPM values were divided by holobiont DPM values to yield the fraction of autotrophically derived carbon shared between partners, which we defined as our metric of host-symbiont cooperation. The degree of symbiont parasitism was calculated as the difference in this proportion of photosynthetically fixed carbon translocated to hosts between heat-treated and paired control samples following the full 17 days of experimental treatment, *sensu Baker et al. (2018)*.

To identify major *Symbiodinium* clades hosted by focal species, DNA was extracted from symbiont fractions of the tissue homogenate for each replicate colony of each species ($n = 60$, all control tank fragments) using Wayne's method (*Wilson et al., 2002*). A restriction digest of the large subunit (LSU) region of *Symbiodinium* rRNA (*Baker & Rowan, 1997*; *Palstra, 2000*) consistently revealed single bands indicating the dominance of a single *Symbiodinium* clade for four of the six species (*A. millepora*, *M. aequituberculata*, *Goniopora columna* and *P. lobata*) whereas communities in *Galaxea astreata* and *Galaxea acrhelia* appeared more variable (Fig. S5). Therefore, DNA was pooled in equal proportions by species ($n = 6$ samples) to identify general species-specific communities using amplicon sequencing of the ITS2 region of *Symbiodinium* rRNA. Additional DNA samples for each of the *Galaxea astreata* and *Galaxea acrhelia* colony replicates ($n = 20$ samples) were also submitted for sequencing at the Genome Sequencing and Analysis Facility at the University of Texas at Austin. Given the high diversity subsequently observed in pooled samples of *P. lobata* and *M. aequituberculata*, additional colony replicate samples of each ($n = 3$ and $n = 5$, respectively) were later submitted for sequencing at Oregon State University's Center for Genome Research and Biocomputing to compare symbiont community composition from individual samples.

## Amplicon sequencing analysis

For sample libraries prepared and sequenced at UT Austin's GSAF, the ITS2 primers of *Pochon et al. (2001)* were used. For colony replicate samples sequenced at OSU's CGRB (individual *P. lobata* and *M. aequituberculata* samples), the ITS2 primers of *LaJeunesse (2002)* were used to prepare libraries. All sequencing libraries were subsequently analysed together. Prior to analysis, raw read data was filtered to remove reads which contained Illumina sequencing adapters or did not begin with the correct ITS2 amplicon primer

sequence using the BBMAP package ver. 37.75 (http://sourceforge.net/projects/bbmap/).
The DADA2 pipeline (*Callahan et al., 2016*) implemented in *R Development Core Team
(2017)* was then used to infer sequence variants. Read data was analysed according to
the following tutorial (https://benjjneb.github.io/dada2/tutorial.html) with the following
modifications: ITS2 primers were trimmed from the beginning of each read and forward
and reverse reads were truncated at 210 and 160 bp, respectively, to remove low quality
bases at the end of reads. Additional paired end reads were discarded if they exhibited
more than one expected error or when a quality score of 2 or less was encountered. In
addition, when inferring sequencing variants using the *dada*() command, the BAND_SIZE
flag was set to 32 as is recommended for ITS data (https://benjjneb.github.io/dada2/
tutorial.html). The distribution of raw read data per sample and reads lost during quality
filtering and processing steps can be found in Table S1. Post-clustering curation of
identified amplicon sequence variants (ASVs) was accomplished with the LULU
algorithm, using the default parameters (*Frøslev et al., 2017*). The MCMC.OTU package
(*Green et al., 2014*) was then used to remove sample outliers with low counts overall
($z$-score < −2.5) and remaining ASVs of abundance less than 0.001 (*Quigley et al., 2014*)
prior to statistical analysis. In total, 12 ASVs were identified across samples that satisfied
these criteria. These high confidence sequence variants were taxonomically classified
through a blast search against the GeoSymbio ITS2 database https://sites.google.com/site/
geosymbio/downloads (*Franklin et al., 2012*), and the best match was recorded. In cases
where variants matched equally well to multiple references, all top hits were reported
(Table S2). Resequencing of the individual *P. lobata* and *M. aequituberculata* samples
indicated that the *P. lobata* pooled sample was contaminated at some stage of the
sequencing process, as individual samples did not match the host pool (Fig. S6). In
addition, two *Galaxea* samples (Gacr1 and Gast4) appeared mislabeled, based on the
pattern of symbiont community differences among species (Fig. S6) and we removed
these samples prior to further analysis. To account for differences between individual
samples and species pools, we calculated the normalized mean abundance of each
variant across individual samples within a species and replaced the pooled sequence
sample with this in silico pooled value, when available.

## Statistical analyses

All statistical analyses were performed in R version 3.4.2 (*R Development Core Team,
2017*). A series of linear mixed models were used to determine the effect of species,
temperature treatment and sampling day on physiological metrics. We used a conservative
approach to evaluate the effect of transmission mode on physiological metrics. Rather
than modeling fixed effects of transmission mode directly, we modeled the fixed effects of
species (levels: Amil, Maeq, Gcol, Plob, Gast, Gacr), sampling day (levels: day 2, day 4,
day 17), temperature treatment (levels: heat, control) and all possible interactions on
*Symbiodinium* cell density, net photosynthesis, and host-symbiont resource sharing using
the *lme* command of the *nlme* package (*Pinheiro et al., 2013*), including source coral
colony identity nested within reef of origin as a scalar random factor. All models were
assessed for normality of residuals and homoscedasticity. Symbiont densities required a

log-transformation to meet normality assumptions. To assess goodness-of-fit, we used the function *rsquared.glmm* to calculate the conditional $R^2$ value for each of our mixed models (*Johnson, 2014*). Significance of fixed factors within models was evaluated using Wald tests and Tukey's post hoc tests were used to evaluate significance among levels within factors and interactions when warranted. In these models, a significant effect of transmission mode would have been detected first as a significant difference among species, but significant differences in the hypothesized direction between each pair of vertical and HTs in the subsequent Tukey's tests were also required to satisfy reporting an effect of transmission mode overall.

To determine the impact of specific *Symbiodinium* community characteristics on responses among species, the DESeq package (*Anders & Huber, 2010*) was used to construct a series of generalized linear models to evaluate differences in in the abundance of each ASV by species with respect to aspects of host-symbiont cooperation: mean initial symbiont density (Day 2, heat samples); the mean fraction of carbon shared by symbionts under ambient conditions (Day 2, 4, 17 control samples), the mean difference in the fraction of carbon shared by symbionts during bleaching (Day17, heat vs. control samples) and the mean difference in symbiont density during bleaching (Day17, heat vs. control samples). Models were run for 30 iterations and for models that did not converge, *p*-values were converted to NAs (the standard notation for missing data), prior to applying a multiple test correction (*Benjamini & Hochberg, 1995*).

Predictive relationships between *Symbiodinium* cell density, the degree of bleaching, cooperation, and symbiont parasitism, in addition to *Symbiodinium* community diversity (quantified using the inverse Simpson index, (*Magurran, 2004*)) and composition (the proportion of Clade D) were explored at the species level using a series of linear models. When a significant difference was detected among species for physiological trait comparisons, the model was re-run within each species and a multiple test correction using the method of *Benjamini & Hochberg (1995)* was applied. Analyses of symbiont community composition were run both using the full six species averages and excluding the species for which individual sample sequencing replicates were not available (*A. millepora* and *Goniopora columna*). As models did not substantially differ, we report results for the full six species.

## RESULTS

### Physiological metrics of thermal tolerance and cooperation

Significant fixed effects of species, temperature treatment, sampling time and the time*treatment interaction were detected for the *Symbiodinium* cell density model ($R^2_{\text{GLMM}} = 0.53$, Table S3). The fixed difference among species was driven by the low symbiont density on average in *P. lobata*, which differed significantly from densities in *A. millepora*, *M. aequituberculata*, *Galaxea astreata* and *Galaxea acrhelia* (Tukey's HSD < 0.05, Fig. 1). No differences were detected in *post hoc* tests between focal species pairs by symbiont transmission mode. Across species, cell densities did not differ between control and heat-treated corals on days 2 and 4, but were reduced in heat treated corals on day 17, by $5.4 \times 10^5$ cells/mg host protein on average (Tukey's HSD < 0.001, Fig. 1).

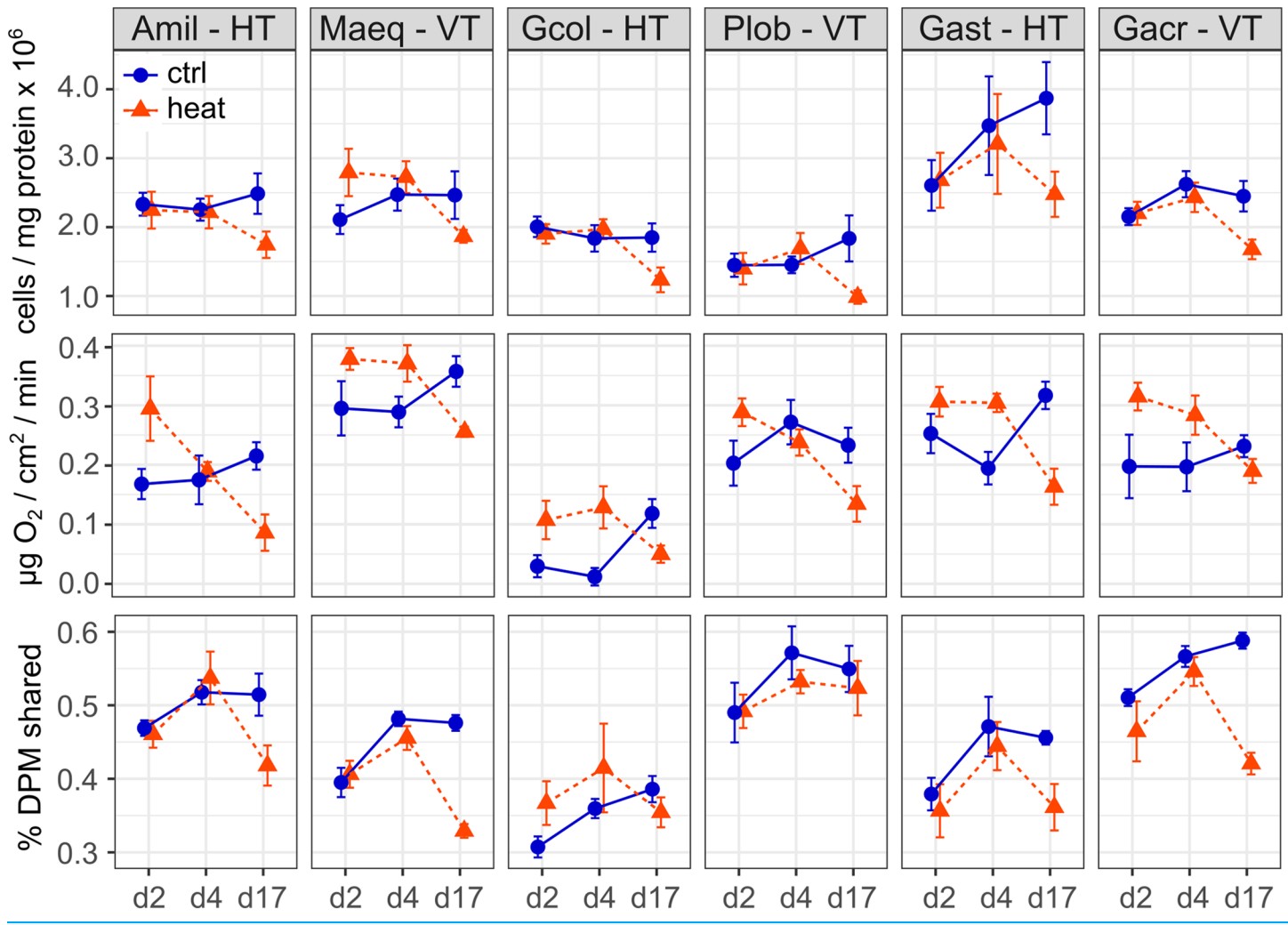

**Figure 1** *Symbiodinium* **cell density, rate of net photosynthesis, and host-symbiont cooperation as a function of experimental treatment and time.** *Symbiodinium* cell density, expressed as cells per mg of host protein, rate of net photosynthesis, expressed as µg $O_2$/cm$^2$/min and the proportion of photosynthetically fixed carbon translocated to the host, calculated as the ratio of total radioactivity in DPM in host to holobiont tissue fractions of focal coral species (Amil: *Acropora millepora*, Maeq: *Montipora aequituberculata*, Gcol: *Goniopora columna*, Plob: *Porites lobata*, Gast: *Galaxea astreata*, Gacr: *Galaxea acrhelia*) representing different symbiont transmission modes (HT, horizontal transmitter, VT, vertical transmitter) under control (27 ℃, blue circles) and elevated (31 ℃, red triangles) temperature following 2, 4 and 17 days of treatment.

The model for changes in the rate of net photosynthesis identified significant effects of species, sampling time and the temperature treatment*time interaction ($R^2_{\mathrm{GLMM}}$ = 0.53, Table S4). Average net $O_2$ production rate was highest in *M. aequituberculata*, differing significantly from *A. millepora*, *Goniopora columna*, *Galaxea acrhelia* and *P. lobata* Tukey's HSD < 0.05, Fig. 1), and lowest in *Goniopora columna*, significantly more so in comparison to *M. aequituberculata*, *P. lobata*, *Galaxea astreata* and *Galaxea acrhelia* (Tukey's HSD < 0.001, Fig. 1). Net $O_2$ production rate was also lower on average in *A. millepora* than in *M. aequituberculata* and *Galaxea astreata* (Tukey's HSD < 0.01, Fig. 1). Considering differences between focal species pairs by symbiont transmission mode, the rate of net photosynthesis was significantly higher in the vertically transmitting

*M. aequituberculata* and *P. lobata* than in their horizontally transmitting counterparts, *A. millepora* and *Goniopora columna*. However, it is important to note that absolute rates of oxygen production among focal species are much lower than previously reported values for corals (*Anthony et al., 2008*). Corals were acclimated to a common light environment, but data on species and time-point specific photosynthesis-irradiance curves are lacking. Consequently, the differences among species may be influenced by variation in species-specific photobiology while differences within species may be influenced by shifting photophysiological performance over the experimental time-course.

Over the course of the experiment, net photosynthesis rates were elevated in heat-treated corals on days 2 and 4 relative to their respective controls (day 2: 0.09 μg $O_2$/cm$^2$/min; day 4: 0.06 μg $O_2$/cm$^2$/min; Tukey's HSD < 0.01, Fig. 1) and reduced in heat-treated corals on day 17, by 0.1 μg $O_2$/cm$^2$/min (Tukey's HSD < 0.001, Fig. 1).

The proportion of carbon photosynthetically fixed by symbionts then translocated to hosts was significantly different among species, temperature treatment, sampling time, the treatment*time interaction and the species*treatment interaction ($R^2_{GLMM}$ = 0.73, Table S5). Carbon sharing in ambient conditions was significantly lower in *Goniopora columna* in comparison to all other species (Tukey's HSD < 0.05, Fig. 1) and highest in the vertically transmitting *Galaxea acrhelia* and *P. lobata*, significantly more so than in *M. aequituberculata* and their horizontally transmitting counterparts, *Galaxea astreata* and *Goniopora columna* (Tukey's HSD < 0.01, Fig. 1). Over time, no significant differences were detected in carbon sharing between control and heat-treated corals on days 2 and 4, but proportional translocation was significantly lower in heat-treated corals on day 17 (Tukey's HSD < 0.001, Fig. 1). Relative to controls, carbon sharing decreased slightly under heat treatment on average in all species save *Goniopora columna*, though the only significant difference was observed in *Galaxea acrhelia* (heat vs. control Tukey's HSD < 0.01, Fig. 1). Although the species*treatment*time interaction was not significant in the final model, this was likely driven by the decrease in heat-treated corals on day 17, which is most evident in *A. millepora*, *M. aequituberculata*, *Galaxea astreata* and *Galaxea acrhelia*.

## Relationships between symbiont density, degree of bleaching and carbon translocation

Symbiont cell density did not explain a significant proportion of the variance in carbon translocation under ambient conditions (control corals on days 2, 4 and 17) due to differences among species ($F_{1,5}$ = 25.57, $P$ < 0.001, Fig. 2). Within species, carbon translocation increased with increasing symbiont density in *Galaxea acrhelia* ($R^2$ = 0.22) but this relationship became non-significant after applying a multiple test correction.

Symbiont cell density in heat-treated corals on day 2 predicted a small portion of the variance in bleaching intensity on day 17 (calculated as the difference in cell density between paired control and heat-treated coral fragments). Greater bleaching was associated with higher initial symbiont cell densities (β = −0.34, $R^2$ = 0.06, $P$ = 0.04, Fig. 3A) and this relationship did not differ significantly among species. A similar, but non-significant trend was also observed between bleaching on day 17 and symbiont cell density on day 4 (β = −0.24, $R^2$ = 0.04, $P$ = 0.07).

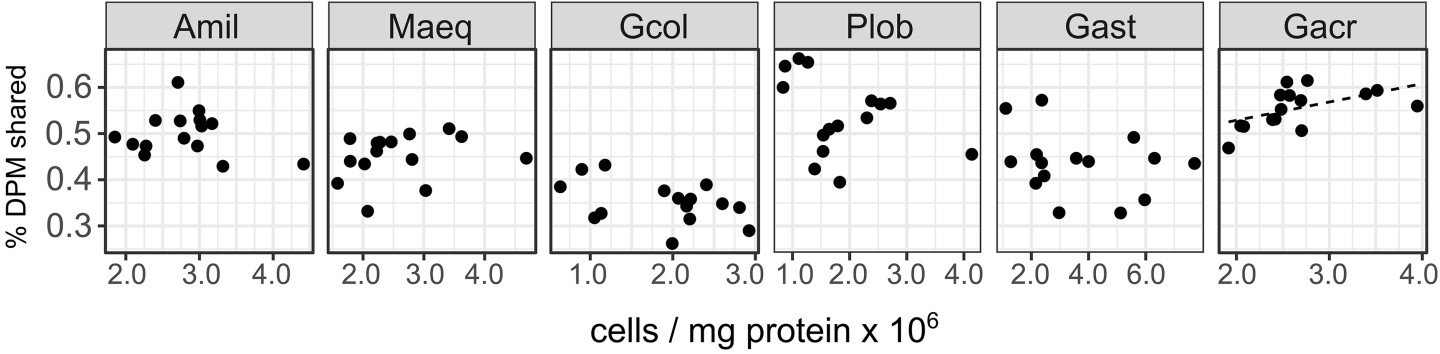

**Figure 2 Host-symbiont cooperation in ambient conditions as a function of *Symbiodinium* cell density across species.** The proportion of photosynthetically fixed carbon translocated to the host in ambient conditions (27 °C) as a function of *Symbiodinium* cell density by species (Amil: *Acropora millepora*, Maeq: *Montipora aequituberculata*, Gcol: *Goniopora columna*, Plob: *Porites lobata*, Gast: *Galaxea astreata*, Gacr: *Galaxea acrhelia*).

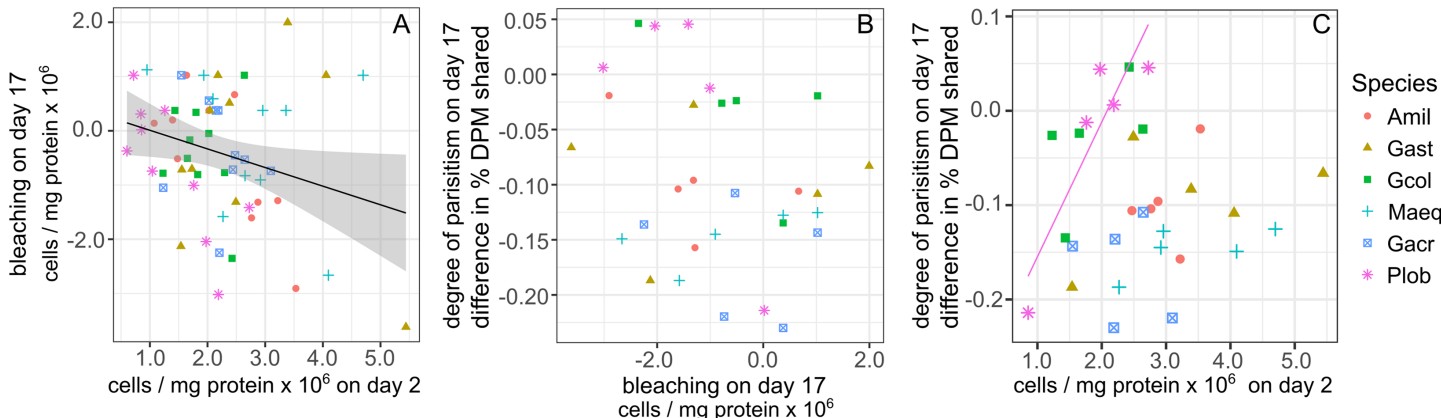

**Figure 3 Relationships between changes in symbiont cell density and the proportion of photosynthetically fixed carbon translocated to hosts.** (A) Bleaching intensity, calculated as the change in symbiont cell density between control (27 °C) and heat-treated (31 °C) samples on day 17 as a function of symbiont cell density in heat treated corals on day 2. (B) The degree of symbiont parasitism, calculated as the difference in the proportion of photosynthetically fixed carbon translocated to hosts between control and heat-treated samples on day 17. (C) The degree of symbiont parasitism as a function of symbiont cell density in heat treated corals on day 2. Regression line shown for Plob only. Amil: *Acropora millepora*, Maeq: *Montipora aequituberculata*, Gcol: *Goniopora columna*, Plob: *Porites lobata*, Gast: *Galaxea astreata*, Gacr: *Galaxea acrhelia*.

No significant relationship was detected between the degree of bleaching and the degree of parasitism (calculated as the difference in carbon translocation between heat and control treated samples of paired corals by source colony) at the end of the experiment (Fig. 3B). Nor was the degree of parasitism explained by differences in symbiont cell density of heat-treated corals on day 2, but relationships differed among species ($F_{1,5} = 6.88$, $P < 0.001$). In *P. lobata*, a higher symbiont density on day 2 was associated with lower parasitism, or an increase in proportional carbon translocation in heat-treated corals on day 17, and this relationship remained marginally significant even after applying a multiple test correction ($R^2 = 0.89$, $P = 0.06$, Fig. 3C).

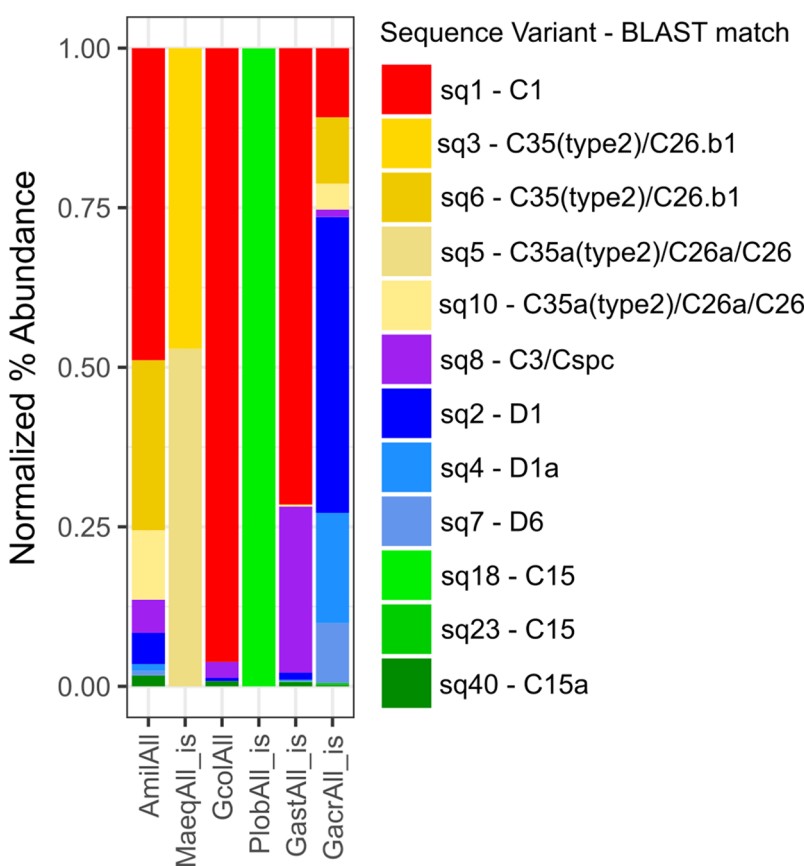

**Figure 4 Normalized percent abundance of ITS-2 sequence variants by pooled sample.** Normalized percent abundance of curated amplicon sequence variants by pooled sample. The 'is' label indicates samples for which sequence variant abundances across individually sequenced samples were pooled in silico. The best BLAST match against the GeoSymbio ITS2 database (*Franklin et al., 2012*) for each sequence variant is also reported.

## The role of *Symbiodinium* community composition

Across all individual and pooled samples, 12 ASVs of sufficient representation (>0.1%) abundance, *sensu* (*Quigley et al., 2014*) were identified and consisted of clade C and D-type *Symbiodinium* (Table S2 and Fig. S6). *Symbiodinium* community composition varied among species (Fig. 4) but no relationships were detected between the abundance of individual ASVs and symbiont density, degree of bleaching or carbon translocation. Nor did we detect any relationship between *Symbiodinium* community diversity overall and symbiont density on day 2, mean carbon translocation under ambient conditions or the degree of bleaching at the end of the experiment (Figs. 5A, 5B and 5D). A marginally significant negative relationship was detected between community diversity and the degree of parasitism at the end of the experiment, where a more diverse community was associated with a greater degree of symbiont parasitism ($R^2 = 0.56$, $P = 0.054$, Fig. 5C). We did not detect any relationships between the percent of Clade D in the symbiont community and symbiont density on day 2, mean carbon translocation under ambient conditions, or the degree of parasitism or bleaching at the end of the experiment

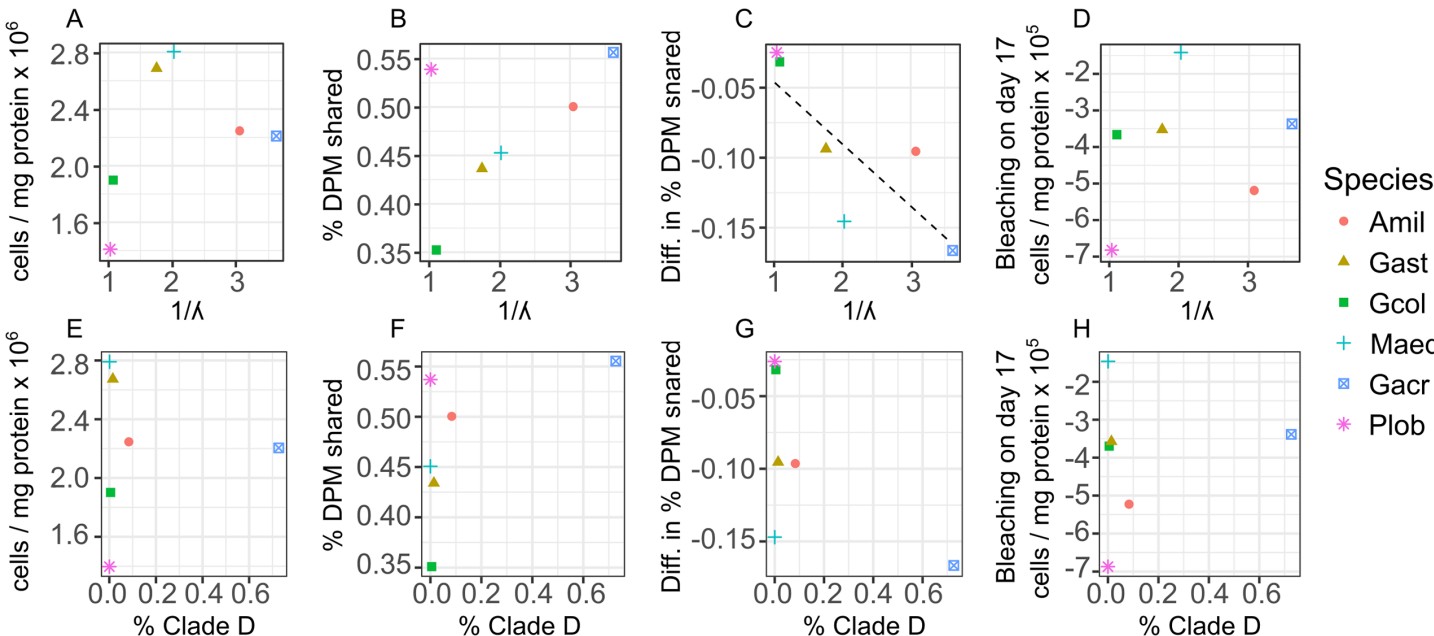

**Figure 5 Relationships between changes in symbiont cell density and host-symbiont cooperation as a function of symbiont community diversity or the normalized proportion of Clade D *Symbiodinium* across species.** Relationships between changes in symbiont cell density or the proportion of photosynthetically fixed carbon translocated to hosts as a function of symbiont community diversity, expressed as the inverse Simpson index (A–D) or the normalized proportion of Clade D *Symbiodinium* (E–H) across species. For the dependent variables: (A and E) mean symbiont cell density in heat-treated corals (31 °C) on day 2; (B and F) mean proportion of photosynthetically fixed carbon translocated to hosts under ambient conditions (27 °C); (C and G) The degree of symbiont parasitism, calculated as the difference in the proportion of photosynthetically fixed carbon translocated to hosts between control and heat-treated corals on day 17.

(Figs. 5E–5H), but this was likely due to the fact that the abundance of D was very low in all species except *Galaxea acrhelia*, however, did exhibit the highest carbon translocation rates under ambient conditions and the greatest transition towards parasitism on day 17 (Figs. 5F and 5G).

## DISCUSSION

Understanding variation in the degree of cooperation between corals and their *Symbiodinium* will be critical for assessing survival potential among species and populations in the face of increasing environmental change (*Lesser, Stat & Gates, 2013*). As in other mutualisms (*Ebert, 2013*; *Frank, 1994*; *Herre et al., 1999*), vertical transmission has been proposed as an evolutionary mechanism for enhancing holobiont fitness in the Cnidarian–algal symbiosis (*Putnam et al., 2012*). However, we did not observe universally consistent differences in cooperation between vertical and horizontally transmitting species. The vertically transmitting *P. lobata* and *Galaxea acrhelia* exhibited the highest levels of carbon translocation in ambient conditions, which we interpret as symbiont-host cooperation, significantly more so than their respective horizontally transmitting counterparts, *Goniopora columna* and *Galaxea astreata*; but cooperation was not different between *M. aequituberculata* and *A. millepora*, and tended to be higher in the latter horizontally transmitting species. However, species-specific photosynthesis-irradiance curves were not measured and the potential for interactions between

photophysiology and baseline rates of carbon translocation must also be explored. We also expected that the degree of breakdown in the host-symbiont relationship under heat stress, which we interpret as a transition towards parasitism *sensu* (*Baker et al., 2018*), would be comparatively intensified in horizontally transmitting species, but again, this was not the case. *P. lobata* did exhibit the least change in carbon translocation in spite of showing the same signs of bleaching as other species, but there was no difference in comparison to *Goniopora columna* which also largely maintained its baseline translocation under elevated temperature. In addition, the greatest relative declines in the proportion of carbon translocated from symbionts to hosts actually occurred in the other two vertically transmitting species, *Galaxea acrhelia* and *M. aequituberculata*.

Although we did not find significant support for the role of vertical transmission we still observed significant differences in cooperation among our six focal species, both baseline differences under ambient conditions and in the degree of transition towards parasitism under elevated temperature stress. We therefore conducted a series of *post hoc* analyses to explore other putative drivers of differential cooperation and thermal tolerance: differences in *Symbiodinium* cell density and/or in symbiont community composition.

The density of symbiont cells was recently proposed as a major driver underpinning the degree of cooperation between coral hosts and symbionts and the functional response of the coral symbiosis to environmental stressors (*Cunning & Baker, 2014*). Studies in other species have shown that corals with greater initial *Symbiodinium* cell densities, as quantified by the symbiont to host cell ratio, are subsequently associated with greater bleaching severity in response to elevated thermal stress (*Cunning & Baker, 2013*; *Silverstein, Cunning & Baker, 2014*). This association has been hypothesized to result from the proportional increase in oxidative cellular stress: more symbionts yield more reactive oxygen species when the photosynthetic machinery is overloaded (*Cunning & Baker, 2014*), though the recent work of *Baker et al. (2018)*, adds another potential explanation. They showed that under non-limiting nutrient conditions, *Symbiodinium* cell division in *Orbicella faveolata* actually increased in response to sub-bleaching temperature exposure, but that the metabolic costs were borne by the coral hosts (*Baker et al., 2018*), supporting the prediction that a transition to parasitism precedes unsustainable proliferation of the symbiont community which ultimately results in bleaching (*Wooldridge, 2009*).

In examining relationships between symbiont cell densities, the intensity of the bleaching response at the end of our 17-day temperature exposure and carbon translocation rates, we did observe a negative relationship between initial symbiont cell density on days 2 and 4 and subsequent bleaching response on day 17, which did not significantly differ among our six focal species. However, initial symbiont cell densities did not predict the degree of the subsequent transition to parasitism. In fact, the majority of species showed a trend in which greater initial cell densities were associated with a greater maintenance of cooperation under bleaching stress. We also found no relationship between the degree of bleaching and the degree of parasitism on day 17, nor did symbiont cell density explain variation in cooperation among species in ambient conditions. For some species, cooperation tended to decrease with increasing density of symbionts, whereas in others it increased. Taken together, these observations support the association between initial symbiont cell density and
subsequent bleaching intensity, but disagree with the proposed role of an alteration of host-symbiont cooperation in mediating the bleaching response.

The discrepancy in results among studies may be due to the importance of nutrient enrichment for observing a parasitic increase in symbiont communities, or in the difference in study duration and sampling design. Corals were exposed to a constant light environment, which likely did not provide a saturating irradiance across species. Consequently, alteration of carbon translocation as a function of species-specific photophysiology may explain baseline differences in ambient conditions, and future work should aim to test this hypothesis. In addition, while corals received supplemental feeding throughout the duration of this experiment, which likely introduced organic nitrogen, we did not explicitly manipulate inorganic nitrogen levels. It is also possible that variation among hosts in their ability to limit their symbionts' nitrogen supply may have influenced the observed variation in the degree of parasitism (*Cunning et al., 2017*; *Wooldridge, 2009*), although additional studies are needed to test this. In addition, *Baker et al. (2018)* exposed corals to a +5 °C temperature ramp over 8 days, sampling once 24 h after the final temperature was reached, whereas the present study increased temperatures by +4 °C over 5 days and maintained that difference for an additional 12 days, sampling at three time points, comparatively earlier and later. We did not observe an initial increase in symbiont cell density or decrease in carbon translocation on days 2 and 4 under elevated temperature, the experimental time-frame most analogous to that of *Baker et al. (2018)*. It is possible that these dynamics occurred during a time frame in which we did not sample; however, our final results argue against this explanation. On day 17, we observed symptoms of bleaching that did not differ across species: symbiont cell densities and rates of net photosynthesis were uniformly decreased. However, the transition to parasitism was not uniform, as some species exhibited significant differences in carbon translocation in response to heat stress whereas others did not. We therefore conclude that while symbiont density alone may be a reasonable predictor of the potential for observing a bleaching response under elevated temperature, it does not predict cooperative dynamics that likely also influence holobiont fitness.

We also explored the role of symbiont community diversity on bleaching stress and cooperation. Predictions regarding the cooperative and fitness benefits of evolving vertical transmission are based on the assumption that the host-symbiont relationship becomes exclusive: symbiont population sizes are substantially reduced, resulting in genetic uniformity, more rapid co-evolution of partner traits and reduction in intra-symbiont community competition (*Herre et al., 1999*; *Maynard Smith & Szathmary, 1995*). In this case, it may not be vertical transmission *per se* that influences host-symbiont cooperation, but the relative diversity of the symbiont community. A prior meta-analysis found that symbiont specificity was correlated with transmission mode, with horizontally transmitting species being more likely to interact with generalist symbionts (*Fabina et al., 2012*). However, the relationship between transmission mode and overall community diversity was not explored. Other more recent studies have also shown the potential for ontogenetic shifts in *Symbiodinium* community composition of putative VTs, potentially indicating the capacity for mixed-mode or cryptic horizontal transmission

(*Byler et al., 2013*; *Reich, Robertson & Goodbody-Gringley, 2017*). Our results show that symbiont diversity does not partition by transmission mode. While communities in the vertically transmitting *M. aequituberculata* and *P. lobata* were largely uniform, consisting predominantly of C35 and C15-type sequence variants, respectively, *Symbiodinium* community diversity was more than twice as high in *Galaxea acrhelia* in comparison to *Galaxea astreata* (1/D = 3.6 vs. 1.7). In addition, community composition in the horizontally transmitting *Goniopora columna* was also highly uniform, second only to that of *P. lobata* (1/D = 1.08 and 1.00, respectively).

In exploring the relationship between symbiont diversity at the ITS2 locus and metrics of host-symbiont cooperation and bleaching independent of transmission mode, we did not find any formally significant correlations, likely due to the fact that our symbiont community analysis was limited to the level of differences among the six species, greatly reducing our statistical power. However, there was a weak negative relationship between community diversity and the degree of parasitism under thermal stress. Species with the most genetically uniform symbiont communities, *P. lobata* and *Goniopora columna*, maintained the highest levels of cooperation in spite of showing signs of bleaching. Yet there was no relationship between community diversity and baseline differences in cooperation among species under ambient conditions, as high rates of translocation were observed in species with both the lowest (*P. lobata*) and highest community diversity (*A. millepora*, *Galaxea acrhelia*), though again, species-specific interactions between photophysiology and carbon translocation remain to be explored.

The presence of particular symbiont types has also been shown to influence holobiont fitness and carbon translocation. For example, conspecific corals hosting clade D *Symbiodinium* exhibit greater thermal tolerance than those hosting C1 or C2-types (*Berkelmans & Van Oppen, 2006*; *Jones et al., 2008*), however, they generally grow more slowly under non-stressful conditions (*Jones & Berkelmans, 2010*; *Little, Van Oppen & Willis, 2004*) and receive less photosynthetically fixed carbon from their symbionts (*Cantin et al., 2009*). We found no significant relationships between the proportional abundance of Clade D-type symbionts and metrics of host-symbiont cooperation or bleaching. Most species had no, or a low proportion of Clade D, but the species with the greatest proportion of Clade D symbionts (*Galaxea acrhelia*) exhibited both the highest carbon translocation under ambient conditions and the greatest transition to parasitism under elevated temperature stress. Similar to our observations regarding symbiont cell density, these results support prior observations that *Symbiodinium* community composition alone is not sufficient to explain variation in holobiont performance (*Abrego et al., 2008*; *Baird et al., 2009a*; *Kenkel et al., 2013*). However, we reiterate that our analyses are limited in comparing only averages among species. There was some variation observed in dominant symbiont types among individual coral colonies within species (Fig. S6) and a priority for future study should be to investigate whether these conclusions hold when considering intraspecific variation in symbiont community composition in addition to these broader interspecific differences.

Quantifying cooperation between symbiotic partners in terms of biologically realistic costs and benefits remains an outstanding question for many symbioses (*Herre et al.,*

*1999*). The transfer of photosynthetically fixed carbon has long been known as a major cooperative benefit to the coral host as up to 95% of a coral's energy requirements can be met through this mechanism (*Muscatine, 1990*); however, reciprocal products shared by hosts with their symbionts remain largely unknown (*Yellowlees, Rees & Leggat, 2008*). Similarly, heterotrophic feeding can offset the need for symbiont-derived carbon in some species and in these cases other symbiont-derived metabolites may be more critical for host fitness (*Grottoli, Rodrigues & Palardy, 2006*). Substantial variation in both intra- and inter-specific bleaching thresholds (*Marshall & Baird, 2000*), suggests that levels of cooperation between host and symbiont may also vary. Significant work has gone into investigating coral bleaching over the past three decades, yet fundamental questions remain unresolved (*Edmunds & Gates, 2003*). Ultimately, a greater understanding of both fine-scale interactions between coral hosts and symbionts and the evolutionary and ecological mechanisms that maintain and strengthen cooperation will be essential for managing these ecosystems (*Davy, Allemand & Weis, 2012*; *Lesser, Stat & Gates, 2013*).

## CONCLUSIONS

This study investigated whether corals employing vertical symbiont transmission also exhibit enhanced cooperation and holobiont fitness. Contrary to theoretical predictions, we did not find significant support for the role of vertical transmission in spite of significant differences in cooperation among our six focal species. In a *post hoc* analysis of other drivers, we found that a greater initial symbiont cell density was associated with a greater bleaching intensity, but this association did not appear to result from an alteration of host-symbiont cooperation. Rather, the reduction in cooperation across species at the onset of bleaching was marginally associated with symbiont community diversity. The theoretical benefits of evolving vertical transmission are based on the underlying assumption that the host-symbiont relationship becomes genetically uniform, thereby reducing competition among symbionts. Taken together, our results suggest that it may not be vertical transmission per se that influences host-symbiont cooperation, but genetic uniformity of the symbiont community, though future work is needed to directly test this hypothesis.

## ACKNOWLEDGEMENTS

The authors gratefully acknowledge the efforts of A Bouriat in maintaining aquaria and processing coral samples. M Salmon, G Milton, A Severati, C Humphrey implemented the experimental aquaria design. Coral collection was accomplished with the help of S Noonan, V Mocellin, A Severati and M Nayfa. P Muir provided advice on coral taxonomic identification. Comments from J Caley, B Schaffelke and multiple anonymous reviewers greatly improved this manuscript.

### Funding

An NSF International Postdoctoral Research Fellowship, DBI-1401165, to Carly D Kenkel and funding from the Australian Institute of Marine Science to Line K. Bay, supported this

work. The funders had no role in study design, data collection and analysis, decision to publish, or preparation of the manuscript.

### Grant Disclosures
The following grant information was disclosed by the authors:
NSF International Postdoctoral Research Fellowship: DBI-1401165.
Australian Institute of Marine Science.

### Competing Interests
The authors declare that they have no competing interests.

### Author Contributions
- Carly D. Kenkel conceived and designed the experiments, performed the experiments, analysed the data, contributed reagents/materials/analysis tools, prepared figures and/or tables, authored or reviewed drafts of the paper, approved the final draft.
- Line K. Bay performed the experiments, contributed reagents/materials/analysis tools, authored or reviewed drafts of the paper, approved the final draft.

### Field Study Permissions
The following information was supplied relating to field study approvals (i.e. approving body and any reference numbers):

Fragments from sixty unique coral colonies, ~20 cm in diameter, were collected from reefs on the Central GBR from the 8–22 April 2015 under the Great Barrier Reef Marine Park Authority permits G12/35236.1 and G14/37318.1.

### DNA Deposition
The following information was supplied regarding the deposition of DNA sequences:

Raw amplicon sequencing data is available at NCBI's SRA: PRJNA338365.

### Data Availability
Kenkel, Carly D., & Bay, Line K. (2018). Exploring mechanisms that affect coral cooperation: symbiont transmission mode, cell density and community composition (Version v1) [Data set]. Zenodo. http://doi.org/10.5281/zenodo.1208684.

### Supplemental Information
Supplemental information for this article can be found online at http://dx.doi.org/10.7717/peerj.6047#supplemental-information.

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
