# Peer review of "Exploring mechanisms that affect coral cooperation: symbiont transmission mode, cell density and community composition"

_PeerJ, doi:10.7717/peerj.6047_

## Round 0.1 · original submission · Major Revisions

Your manuscript has now been reviewed by three experts in the narrow field of symbiont transmission biology of corals. The reviews were thorough and professional and raise a number of issues with the manuscript that should be addressed in a thorough revision for reconsideration. Two of the reviewers were complimentary of the work, suggesting that with attention to the recommendations of all three reviewers the manuscript can be accepted to PeerJ.

As Editor I wish to stress that there was considerable disagreement among the reviewers, and I concur with most of the points raised. I must warn you that because of the magnitude and scope of some of the criticisms there is every likelihood that unless the reviewers are satisfied by the scope of the revisions the manuscript may eventually be rejected. In particular there are several key areas where I recommend you consider how you can revise the manuscript to improve interpretability and longevity in the context of other studies on corals:

ITS amplicon sequence variant bioinformatics should be handled more consistently. The sequencing of two different amplicon libraries is problematic already, but combining different primers makes this very rough, and essentially prevents the use of ASVs as taxonomic units without specific justification: I concur with reviewers that more coarse clustering approaches are warranted with such a chimeric amplicon dataset. Also the different sequencing sets should be used as a statistical control in all models, which may prevent some comparisons. If the conclusions are robust the variation in sequencing approaches will not matter, but this must be carefully justified.

Clearly presenting the methods used, caveats, and calculations used for derived variables is critical for all scientific publications. I suspect these issues can be cleared up with a rigorous revision and attention to detail.

Rates and stocks should be presented within the context of established studies where the focus was on accuracy in measurement. This is crucial for interpretation of pattern when the results are couched in the language of environmental biology where exact numbers matter, not just relative changes. I concur with R2 on the lack of work done to frame the photosynthetic rates with other measured values for related species, but this is just one example.

All data and R code and bioinformatic scripts used should be presented clearly…there seems to be disagreement among the reviewers on the openness of the analyses. Also, please consider moving some of the key methods back into the paper where the reviewers were frustrated by inaccessibility.

One issue that may be insurmountable in the experimental design is the inconsistency in where and when the corals were collected. These factors should be accounted for statistically where possible, and if covariation between environmental variation and biological responses is not able to be orthogonalized then some of the conclusions must be removed.

We look forward to your thoroughly revised manuscript and point-by-point response to reviewers.

Reviewer 1 ·

Basic reporting

Writing is excellent. Ideas are well introduced and contextualized with the literature. Figures and tables are clearly presented.

Experimental design

Study is well designed and statistical analyses are rigorous.

Validity of the findings

I have two comments which may impact the results and interpretation:
1) The constraint that Symbiodinium sequence variants were only retained if present in >3 samples may exclude too much data, especially since some species were only comprised of a single sample (e.g., Amil). Therefore, sequence variants specific to Amil may be excluded. This should be corrected or the approach justified.
2) There appears to be variation in the dominant symbiont among colonies of Gast. If these are separated in data analysis (or this variation accounted for statistically, i.e. with dominant symbiont added as an additional predictor), might some of these results change? This needs to be addressed in some way. There is also the possibility that A. millepora samples contained different dominant symbionts, but since samples were not sequenced individually this is difficult to test. The gels provide some indication that they are all C-dominated and that D is present at lower levels, but this could be addressed also in the text.

Additional comments

This study investigates whether a coral’s mode of symbiont transmission influences the degree of ‘cooperation’ (measured as proportion of fixed carbon translocated) and bleaching in response to thermal stress. While the transmission mode was not a significant predictor of these traits, the authors explore how they are related to both symbiont density and diversity. With these data, the study supports a relationship between high symbiont density and bleaching susceptibility, but does not support a shift to parasitism among symbionts as a trigger of the bleaching response. The paper is very well written, and the results are presented clearly and contextualized well with existing literature. I commend the authors for conducting rigorous statistical analyses and making them reproducible by providing all data and R code.


Abstract
-Briefly define vertical/horizontal transmission in Abstract for readers unfamiliar with the terminology

Materials and Methods
-Were all corals collected from the same reef? How close together? Was depth the same? (i.e., were these corals likely acclimated to similar environmental conditions?). What was the temperature at the time of collection? I see now some of this information is in the Supplement, but important bits could be brought into the main text. The depth indicated is less than 10m, leaving room for significant potential variation among species?
-What type of lights were used? Was light ramped up and down each day? Light levels are relatively low in the treatment tanks.
-Line 135: Change Waltz to Walz
-Line 137: Interpretation of declining effective quantum yield as onset of bleaching response?
-Line 153: Change holobiont to symbiont?
-Line 170: add “sequencing” between “for” and “at”
-Line 195: Since samples from individual corals were pooled by species, does the removal of sequence variants appearing in less than three unique samples mean that they had to appear in at least three unique species? If so, this seems inappropriate as many species-specific sequence variants may be expected. Since for some species additional individual samples were sequenced, does this cutoff favor the retention of sequence variants found in these species?

Results:
-Line 245-246: Here a non-significant effect of transmission mode on symbiont density is reported. However, the statistical analysis of transmission mode as a predictor is not described in the data analysis section. Was a separate model fitted replacing the species predictor with transmission mode?
-Figure 1: It would be helpful to include ‘HT’ or ‘VT’ after each species name in the facet titles to remind the readers which species are horizontal or vertical transmitters... Picky point: the abbreviation for M. aequituberculata should probably be ‘Maeq’ instead of ‘Maqe’, and likewise G. acrhelia should probably be ‘Gacr’ instead of ‘Gach’.
-Line 304: Change Fig. 4A to Fig. 4? Doesn’t appear to have multiple panels...
-Line 307: Add references to Fig. 5A, Fig. 5B, Fig. 5D.

Discussion:
-Line 373: I would suggest adding another sentence or two expanding on the idea that nutrient enrichment may be necessary to observe a shift to parasitism, i.e. a decrease in the proportion of fixed carbon shared with the host. Indeed, I think this idea is a key component of the Baker et al. 2018 findings. If symbionts are able to access more nitrogen, they may be able to retain more of their fixed carbon to build biomass, thereby becoming more ‘parasitic’. Nutrient enrichment therefore favors this scenario, and the variation among hosts observed here could potentially be explained by variation in the ability of the host to limit the symbionts’ access to nitrogen, as also discussed by Wooldridge (2009 Mar Fresh Res) and Cunning et al. (2017 J Theor Biol).


Supplementary Materials:
Figure S5: Could the colors be labeled with the closest blast hit instead of the OTU id? This figure indicates that within some species (Gast), variation in the dominant symbiont existed among colonies – how could this have influenced the results? Would separating these samples by dominant symbiont be valuable and potentially reveal trends that are masked by pooling these samples? Some of the samples presented here were removed from data analysis as indicated in the Methods section – could this also be indicated somehow in the figure to remind the reader of this?

Reviewer 2 ·

Basic reporting

1. All the derived variables lack clear justification and calculation information in the methods. The entire paper is set on the premise of transmission strategy, which is never actually tested.
2. How are these physiological measures related to fitness and cooperation? The authors are discussing the results as fitness and cooperation, but have measured proxies without a clearly articulated linkage. They also have derived variables that are not clear, such as degree of symbiont parasitism.
a. fitness and cooperation - need to define and justify the proxies used for this upon first or early mention (e.g., Line 97), given that these are reported as primary results in lines 111-118 and cannot be interpreted without further detail. These lines cannot be interpreted without further information.
3. The discussion is much more of what the introduction should be. The authors are not testing vertical transmission, so why not just contextualize the paper with the hypotheses of the derived variables, given they are well defined and justified at the start.

Experimental design

1. All pairs of species were collected from different reefs at different times. Thus all pairs had different thermal histories and acclimation conditions. This should be discussed.

2. Methods for respirometry are not fully detailed and result in problematic data. There is no indication of oxygen was measured at the initiation of the experiment. No light or temperature data are provided. No mixing or stirring information is provided. It is not clear if the rates accounted for the volumes of the chambers and differential displacement due to nubbin and stand sizes. There is no information on if the water was filtered. The authors cannot count on the supplementary data of Strahl et al to make their case for the single endpoint measurements used in the current study. This is all apparent in the photosynthesis data themselves, where the rates seem to be an order of magnitude from the cited protocol (Strahl et al 2015) and at least an order of magnitude off from the literature in general (e.g., Anthony et al 2008 www.pnas.org cgi doi 10.1073 pnas.0804478105).


3. If protein was measured using the BCA kit following manufacturers instructions, then the measure that is reported is not total protein, but soluble protein. This can problematic to normalize to, as the BCA kit can also quantify lipids and lipoproteins, which would likely differ by species. I would recommend also normalizing cell density to surface area to check if patterns remain the same.

4. Sequencing was done very haphazardly with 2 different primer sets, at 2 different facilities, with samples pooled across all replicates, and contamination and mislabeling in some samples, reducing the confidence in the data. It is unclear when the samples were collected in the experiment and how the community at that single time point relates to the response variables measured at different time points. This confounds the statistical tests.

5. Line 139: This is an oversimplification of EQY and not always indicative of subsequent bleaching.

Validity of the findings

1. The major hypothesis was not actually tested statistically. The authors state “We compared physiological components of cooperation and fitness between horizontal and vertical transmitters in a phylogenetically controlled design using three pairs of related coral species exhibiting different strategies.” They however only used “Species” as a main factor, which ignores the factor of symbiont transmission that the experiment was supposedly designed to test. With this approach they are essentially saying species differ. The authors report “No differences were detected between focal species pairs by symbiont transmission mode.” But this analysis was not included anywhere. Similarly lines 257-259.

2. There are a multiple references to derived variables that are not clear in calculation or definition. For example the authors need to define degree of parasitism and bleaching intensity, cooperation, fitness, etc clearly with rationale.

3. Every response variable has some issues with it. Respirometry methods are not up to standard and rely on others data. Cell density data normalizations are not Total protein. Sequence data are a grab bag of approaches and represent pools of unknown timepoint. Derived variables are not clearly defined in terms of calculation or rationale for the calculation.

4. No raw data or analytical scrips from the study are provided, only references.

5. Due to all of these factors, it is not possible to judge the veracity or validity of the authors’ conclusions.

Additional comments

The authors have examined several physiological responses of corals to thermal stress in the context of exploring mechanisms that contribute to the symbiotic function. While the hypothesis that vertical transmission contributes to enhanced cooperation and fitness is solid and a timely issue, the work does not actually appear to test this hypothesis. Primarily, there are issues with the methodologies of all response variables that generate issues with data quality and capacity to compare statistically. Further, there is a lack of clarity in the terminology and derived variables used for additional analyses and a lack of all physiological data and analytical scripts.

Other general comments:
line 53: replace suggesting with indicating
Line 91: clarify species of what, coral-Symbiodinium pairs?
Line 96: Baird et al 2009 more comprehensive ref (Annu. Rev. Ecol. Evol. Syst. 2009. 40:551–71)
Line 128: PAR units are unclear. Also what is meant by peaking at 130-160? Was this a fluctuating light cycle? Include light data in supplement.
Line 146: substitute “fraction of autotrophically derived carbon” for resource
Line 152: separated
Line 154: What is the need for skeletal volume? Measured, but not used again in manuscript.
Line 154: This is not total protein, but soluble protein if only the manufacturer’s instructions were followed.
Line 198: Include reference database a supplementary file.
Line 221: Now have switched to calling responses thermal tolerance and cooperation, but have still not defined them
Lines 232: Clarify how exactly each of these were calculated to include in the statistical analysis. Also check the sentence for missing information.
Lines 276-278: why was the full interaction not in the final model? The stats section Lines 223 indicates all possible interactions were included.
Supp Mat Lines 8-14 include collection dates for each reef location.

Reviewer 3 ·

Basic reporting

The manuscript is generally well written and follows a construction that adheres to the typical formatting guidelines required of original research. Raw data and the scripts to reproduce the outputs are all provided which is excellent. I agree with the overall premise and intent of the manuscript and it is clear that symbiotic corals are undoubtedly important model systems for the study of host:symbiont coevolution and the spectrum of mutualism:parasitism (although your assumption regarding the tendency towards specificity/fidelity in vertically transmitting species is questionable given recent studies showing cryptic horizontal transmission in vertically transmitting coral species: starting with Byler et al 2013 PLOS ONE and more recent examples, such as Reich et al 2017 PLOS ONE). Indeed, considering the global degradation of coral reefs and research initiatives such as the progressive management of coral reefs (restoration, guided evolution, etc), research themes such as presented here, primarily related to the functional biology of coral/Symbiodinium species combinations under stress, are more timely than ever. However, the manuscript could have been presented better and more care must be taken during the preparation and process of submission. For example, a number of cited references are missing from the reference list (e.g. Green et al 2014), there are multiple instances where the formatting of genus/species names is incorrect (e.g. L189, L424), informal abbreviation is used without explanation (e.g. “ctrl” and “NAs”, nit-picky but it is important to be explicit), and important parts of the methodology are needlessly moved to the supplementary sections. In addition to these minor problems, I have several major concerns with how the data have been collected and analysed that likely preclude the publication of this text in its current form.

Experimental design

I have few issues with the design or replication overall (although it is unfortunate that a number of samples had to be pooled). My primary concern is with the processing of sequencing data, which I describe in the sections below. The methods used to assess symbiont diversity are simply not rigorous enough.

Validity of the findings

The most significant problem with this study is the analyzing of the ITS2 amplicon data. The authors have used an amplicon sequence variant (ASV) approach, rather than the well established OTU approach to analyse the ITS2 region (although confusingly the R scripts in the supplementary files contain references to “OTUs"? Recycling of old scripts?). While the case for the usefulness of ASVs has been made (Callahan et al 2017 ISME), it has been quantitatively demonstrated that ITS2 amplicon sequencing massively over-inflates diversity estimates for Symbiodinium. A significant aim of this study is to “investigate whether symbiont community composition could explain patterns of fitness and cooperation among horizontal and vertical transmitters at a species level”. Furthermore, the primary conclusions in the closing paragraphs of this manuscript are centered around relationships between cooperation and Symbiodinium ‘diversity’. These are important themes both in relation to the fundamental biology of the coevolution of host and symbiont(s) and also for researchers who need to strategize/prioritise certain hosts/symbiont species combinations in the anticipative management of corals reefs. However, it is widely known and recognised that the ITS2 marker is multicopy in nature and single genomes are comprised of ITS2 variants that are in various stages of homogenization due to convergent evolution (See Smith et al 2017 ISME for an ITS2 + psbAncr approach). The distances between intragenomic ITS2 variants will likely vary lineage to lineage, but if we take an example from the present study, Symbiodinium ITS2 type C1 (S. goreaui), Arif et al 2011 (Molecular Ecology) found that amplicon sequencing of a monoclonal C1 cell line (rt152) returns a multitude of intragenomic ITS2 variants. This is completely ignored by the present study. It was found that the vast majority of these C1 intragenomic variants can be collapsed into a single biologically meaningful “OTU” using the 97% similarity threshold. The approach in the present study finds up to eight ITS2 type C1 amplicon sequence variants (ASV) that may simply represent intragenomic variants of a single species. If we assume that this is at all possible, including Symbiodinium diversity as a factor in your mixed effect model makes little sense and simply does not evaluate the proposed aim. Diversity and performance/function are not reconciled in any biologically meaningful way. I would suspect that using the suggested clustering approach of Arif et al will collapse all of the presented diversity down to < 10 OTUs. At the very least these data must be explored using this approach in tandem with the present analysis if the goal of comparing diversity as a function of cooperation is to remain as part of the study.

Regarding the physiological measurements.

Please clarify at what time the “peak irradiance” occurs (line 118)? Photosynthetic parameters such as fluorescence transients respond strongly to light but photochemistry also follows endogenous rhythms (i.e. including non-photochemical quenching which is where much of the drop in EQY comes from during the day).

For the photosynthesis rate measurements, were the corals removed from their aragonite plugs before moving them to the incubator chambers? If not, did the “blank” incubations contain coral-free aragonite plugs? From experience the aragonite plugs quickly become covered in crustose coralline algae which also will produce a significant amount of O2 during the incubation. This would need to be accounted for.

---

## Round 0.2 · Major Revisions

The reservations expressed by Reviewer 2 are all clear and indicate significant shortcomings in the validity of the findings as expressed. I am recommending major revisions, and request that the authors write a detailed and specific response to the points raised by R2 in both the first and this current review. In revision, I request that the authors temper their conclusions in the abstract, introduction and discussion regarding these points raised by R2 about rates of photosynthesis and respiration, and qualify the issues with the methodology up front in both the results and discussion. In essence the goal is to ensure that the paper is clear about the limitations of the methodology in reaching the conclusions presented.

Reviewer 1 ·

Basic reporting

no comment

Experimental design

no comment

Validity of the findings

no comment

Additional comments

My concerns have been addressed in this careful revision. I commend the authors on a solid, transparent dataset and analysis.

Suggestions:

Line 30: Replace “metrics of fitness” with “performance”

It would be helpful to more explicitly define “cooperation” in the abstract. The sentence beginning on line 28 implies the meaning of cooperation as translocation of photosynthate, but there are quite a few words in between “translocation” and “cooperation”, so this connection does not immediately jump out at the reader. I suggest restructuring this sentence to make this clearer, and/or including “(i.e., translocation of photosynthate)” after “cooperation” on line 32.

Reviewer 2 ·

Basic reporting

Fitness is used in many different ways throughout the analysis. Consistency and discretion is necessary as actual fitness is not measured and fitness and cooperation are used synonymously at points and not others. Address throughout the manuscript.
e.g., Line 281: remove the word fitness here

Raw data and code are now shared in supplement.

Experimental design

Statistical design is still not clearly rationalized.

If testing the hypothesis of transmission mode and specifically choosing 3 “replicates” of each mode, provide rationale as how it is conservative to use species to test this in the analysis, which inflates the df available for the test.
Lines 277-280

Validity of the findings

The respirometry data have severe issues that are not addressed by the revision and cannot be addressed by simply stating light levels were lower.


1) PI curves were not completed in the current study and it is assumed the same light levels apply in the same way to multiple coral species from multiple reefs. Therefore measurements were done most likely as the authors indicate at pre-saturating irradiance and at different places on different species PI curves, which means they are potentially comparing apples to oranges. It is precisely because “phosynthesis-irradiance curves are logarithmic, showing a marked increase between 0-500 μmol quanta m-2 s-1, but plateauing at higher irradiance levels.” that the authors should have done their own PI curves and chosen to work at a saturating irradiance for all their species. Their data would then be comparative and more likely would match that reported in the literature from values measured at Pmax.

2) These light and photosynthesis issues (lack of comparison at Pmax) then carry over to the quantification of photosynthetically fixed carbon translocated to the host and this needs to be discussed as it has implications for the entire discussion/conclusions.

3) The two-point method described and validated in Strahl et al. (Strahl et al. 2015) apply only to the conditions (physical, biological, and experimental) of Strahl et al. 2015. This validation should also have been done in the current study. Also it is a two point method n Strahl et al, not an end-point method.

4) The photosynthesis values may be calculated incorrectly. It looks like the volume of the chamber was only accounted for 1 time. It appears the authors may need to convert both blank and sample to mg by multiplying by their respective volumes prior to subtracting them.

5) The photosynthesis measurements appear to be gross, not net. Dark respiration measurements were not reported.

Additional comments

The authors have responded positively and in detail in review, but some issues/weaknesses remain that I am not sure can be addressed without new measurements. As it is unlikely that new measurements can be made at this time, it is critical that the authors remove the data or place strong caveats clearly up front with regards to the respirometry data.

Other Comments:
Line 97 – fitness proxies
Line 114 – proxy for holobiont fitness
Lin 166 -171 – place caveat here for single endpoint measurement of rates
Line 176 – what are the light conditions here? It is not clear to say identical to the experimental conditions as light was fluctuating and would have changed over the 5 hour incubations. Add what time of day these were done and the average light level estimate for this time and clarify that this is an estimate.
Line 182: host protein (as assessed below)
Line 293: include citation for inverse Simpson index, or otherwise clarify

Reviewer 3 ·

Basic reporting

The authors have provided satisfactory responses to my comments and I recommend this manuscript for publication.

Experimental design

The authors have provided satisfactory responses to my comments and I recommend this manuscript for publication.

Validity of the findings

The authors have provided satisfactory responses to my comments and I recommend this manuscript for publication.

Additional comments

The authors have provided satisfactory responses to my comments and I recommend this manuscript for publication.

---

## Round 0.3 · Minor Revisions

Please make every effort to clarify the shortcomings of your experimental design, particularly with the reviewers issues with respect to control of light. I agree with the reviewer that it is critical that readers are aware of the importance of considering light levels (among a host of other variables) in the design of experiments, and while this does not devalue your results it is important to be very clear about the limitations and constraints placed on the results because of the light levels used.

Once you have addressed the suggestions of the reviewer adequately I believe this manuscript will be accepted, but I cannot guarantee this.

Reviewer 2 ·

Basic reporting

The basic reporting is clear

Experimental design

Point 3 in response

The light level issues are not fully addressed by looking over time, within a species. PI curves can change through time under thermal stress and for accurate rigorous results PI curves should have been run prior to all time points to determine if the curves have shifted and light levels were within the saturating part of the curve and not sub-saturating or inducing photoinhibition. The authors should also include this caveat in their tests of the onset of the bleaching response and clarify that they are lacking for this study and should be study specific. E.g., “but data on species-specific photosynthesis-irradiance curves are lacking in this study. Consequently, these differences among species may be influenced by variation in species-specific photobiology and differences within species may be influenced by shifting photophysiological performance within the experiments."

Also this argument of testing within species through time is counter to the approach or conclusions between species or transmission mode.


Point 4 in response

The light levels absolutely impact carbon production and thus carbon translocation potential. This should be stated.

Validity of the findings

The validity of the findings are impacted by the methods, specifically light levels.

---

## Round 0.4 · accepted · Accept

Thanks for your attention to the recommendations of the reviewers. We trust this manuscript will be widely appreciated in the field.

#